# Altered expression of genes regulating inflammation and synaptogenesis during regrowth of afferent neurons to cochlear hair cells

Chen-Chi Wu[1,2☯], Aurore Brugeaud[1,2☯], Richard Seist[1,2,3], Hsiao-Chun Lin[1,2], Wei-Hsi Yeh[1,2,4], Marco Petrillo[1,2], Giovanni Coppola[5], Albert S. B. Edge[1,2,4‡]*, Konstantina M. Stankovic[1,2,4,6‡]*

1 Eaton Peabody Laboratories and Department of Otolaryngology—Head and Neck Surgery, Massachusetts Eye and Ear, Boston, Massachusetts, United States of America, 2 Department of Otolaryngology—Head and Neck Surgery, Harvard Medical School, Boston, Massachusetts, United States of America, 3 Department of Otorhinolaryngology-Head and Neck Surgery, Paracelsus Medical University, Salzburg, Austria, 4 Program in Speech and Hearing Bioscience and Technology, Harvard Medical School, Boston, Massachusetts, United States of America, 5 Program in Neurogenetics, Department of Neurology, University of California Los Angeles, Los Angeles, California, United States of America, 6 Harvard Program in Therapeutic Science, Harvard Medical School, Boston, Massachusetts, United States of America

☯ These authors contributed equally to this work.
‡ These authors jointly supervised the project and contributed equally to this work.
* konstantina_stankovic@meei.harvard.edu (KMS); albert_edge@meei.harvard.edu (ASBE)

**Data Availability Statement:** The data are available from NCBI GEO (accession number GSE130495).

## Abstract

The spiral ganglion neurons constitute the primary connection between auditory hair cells and the brain. The spiral ganglion afferent fibers and their synapse with hair cells do not regenerate to any significant degree in adult mammalian ears after damage. We have investigated gene expression changes after kainate-induced disruption of the synapses in a neonatal cochlear explant model in which peripheral fibers and the afferent synapse do regenerate. We compared gene expression early after damage, during regeneration of the fibers and synapses, and after completion of *in vitro* regeneration. These analyses revealed a total of 2.5% differentially regulated transcripts (588 out of 24,000) based on a threshold of p<0.005. Inflammatory response genes as well as genes involved in regeneration of neural circuits were upregulated in the spiral ganglion neurons and organ of Corti, where the hair cells reside. Prominent genes upregulated at several time points included genes with roles in neurogenesis (*Elavl4* and *Sox21*), neural outgrowth (*Ntrk3* and *Ppp1r1c*), axonal guidance (*Rgmb* and *Sema7a*), synaptogenesis (*Nlgn2* and *Psd2*), and synaptic vesicular function (*Syt8* and *Syn1*). Immunohistochemical and in situ hybridization analysis of genes that had not previously been described in the cochlea confirmed their cochlear expression. The time course of expression of these genes suggests that kainate treatment resulted in a two-phase response in spiral ganglion neurons: an acute response consistent with inflammation, followed by an upregulation of neural regeneration genes. Identification of the genes activated during regeneration of these fibers suggests candidates that could be targeted to enhance regeneration in adult ears.

**Funding:** This work was supported by grants from the Department of Defense W81XWH-15-1-0472 (KMS), National Institute on Deafness and Other Communication Disorders R01 DC007174 (ASBE) and R01DC015824 (KMS), the Barnes Foundation (ASBE and KMS), Nancy Sayles Day Foundation (KMS), the Zwanziger foundation (KMS), the National Science Council of the Executive Yuan of Taiwan, 100-2314-B-002 -040 -MY3 (CCW), and by Sheldon and Dorothea Buckler.

**Competing interests:** Dr. Edge acknowledges a competing interest as a cofounder and consultant to Decibel Therapeutics and Audion Therapeutics. This does not alter our adherence to PLOS ONE policies on sharing data and materials.

## Introduction

Regeneration of auditory neurons and their peripheral fibers would be clinically significant because auditory nerve and hair cell synaptic dysfunction often accompanies hearing loss. The primary afferent neurons of the auditory system are postsynaptic to sensory hair cells. The inability of peripheral fibers in the auditory system to undergo synaptogenesis in the adult is thought to be a significant cause of sensorineural hearing loss. Retraction of fibers is normally followed by neural degeneration, in which neurons are lost after the fibers retract beyond the non-myelinated region to the spiral ganglion neuron cell bodies in Rosenthal's canal [1–3]. The loss of synapses in noise-induced models of hearing loss is rapid and is not reversible [4]. Peripheral fiber retraction has been suggested to be reversible in rats after similar damage [5], but quantitative studies of the fate of spiral ganglion neurons after noise-induced loss of terminals showed no evidence for synaptic recovery in mature animals [1]. This is the only peripheral nerve that has no ability to regenerate.

An exception to the lack of regeneration occurs in neonatal ears, in models in which the spiral ganglion fibers regrow to the hair cells and form new ribbon synapses *in vitro*. This has been demonstrated by the addition of spiral ganglion neurons to cultures of the "de-afferented" organ of Corti in which the spiral ganglion neurons have been cut or poisoned. In these models, we have shown that the synapses are reformed [6]. We have also shown that inhibition of guidance molecules such as repulsive guidance molecule a (RGMa) [7] and neurotrophins [6] promotes new synapses with hair cells, whereas a lack of glutamate at the synapse decreases afferent synaptogenesis [6]. In this paper, we use a model of glutamate-toxicity *ex vivo* in which the glutamatergic hair cells and the afferent neurons are bathed in kainate. Kainate, a neuroexcitatory glutamate analogue that activates glutamate receptors, is used to model glutamate excitotoxicity *in vivo* [8] and *in vitro* [9]. Glutamate is an excitatory neurotransmitter at the synapse between the hair cells and auditory nerve fibers [10, 11]. Excessive glutamate or its analogues damage neurons expressing glutamate receptors. In the inner ear, excitotoxicity has been implicated in noise-induced [12, 13] and age-related [14] hearing loss, and other cochlear pathologies that are related to ischemic or anoxic events [15]. In the model we used in this study, derived from a newborn animal, the fibers regrow and form synapses with hair cells. We quantified the extent of reinnervation to approximately 60–70% (Fig 1D). Comparison to *in vivo* models of glutamate toxicity, including noise damage, would suggest that the primary damage occurs at the postsynaptic site. Hair cells may also be affected in noise damage and the lesion may be more complex. Mouse neonatal cochleae on postnatal day 4, the age of dissection, are at a late developmental stage at which hair cell innervation is refined with neurite retraction and synapse pruning, before the onset of hearing around P10 [16, 17]. The purpose of this study is to better understand the genes involved in regeneration in the newborn for comparison to the adult. Genes that are no longer expressed in the adult may account for the loss of regenerative capacity, and the molecular mechanisms underlying the neuronal regeneration remain unclear. The current study begins to characterize the mechanism of neuronal regeneration by bioinformatic analysis of differentially expressed genes.

## Materials and methods

All experimental procedures were approved by the Institutional Animal Care and Use Committee of Massachusetts Eye and Ear and conducted in accordance with the NIH Guide for the Care and Use of Laboratory Animals.

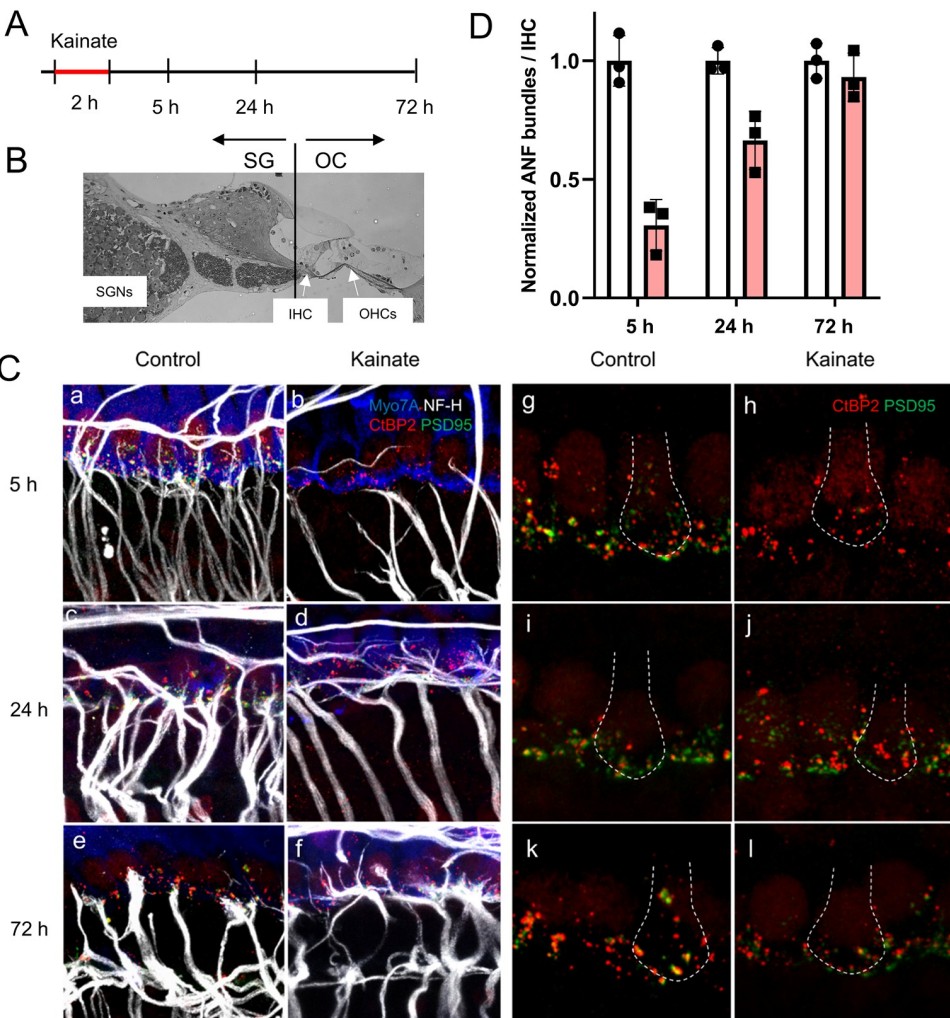

**Fig 1. Synaptic regeneration in cochlear explants after kainate treatment.** (A) Timeline of experimental procedure. (B) A cochlear cross section showing the plane of microdissection separating the spiral ganglion (SG) fraction from the Organ of Corti (OC) fraction. (C) The afferent nerve fibers of spiral ganglion neurons, seen with immunostaining for neurofilament (in white), have synapses with hair cells, as revealed by immunostaining for PSD-95 (in green) and CtBP2 (in red) under each hair cell, stained with Myo7A (in blue), after 5 h in culture (Control). Treatment of the mouse cochlea with kainate for 2 hours resulted in massive loss of the peripheral processes and post-synaptic densities of spiral ganglion neurons by 5 h (Kainate). By 24 h, the peripheral processes had grown and formed synapses (characterized by juxtaposition of CtBP2 and PSD-95 immunostaining) with hair cells (Kainate). At 72 h, there was fiber loss in both the control and treated (Kainate) cultures. Panels a-f highlight auditory nerve fibers. Panels g-l highlight synapses; the white dotted line in each panel outlines a hair cell. Scale bars 20 μm. (D) Quantification of auditory nerve fiber (ANF) bundles per inner hair cell after kainate treatment (red bars) normalized to untreated samples (white bars) at the same time point. Data are presented as means ± SD per group. N = 3 explants per group.

## Cochlear explants and kainate treatment

Cochleae were dissected using sterile conditions under a Zeiss Stemi 2000 dissection microscope. Organotypic explant cultures were prepared from the cochleae of 3 to 5 day old postnatal C57BL/6 mice [18, 19]. The heads were bisected mid-sagittally, the cochleae were removed and placed in ice cold Hank's balanced salt solution (HBSS) (Invitrogen). Cochlear explants, containing the organ of Corti and spiral ganglion neurons, were gently freed from the otic capsule and spiral ligament. The explants were transferred using a wide-mouth pipette containing

a small amount of HBSS from the dissection dish into a 4-well dish (Greiner Labortechnik) coated with fibronectin (BD Bioscience). The tissue was oriented so that the apical surfaces of the hair cells were pointing up and the basilar membrane was directed toward the fibronectin substrate. Excess medium was removed by aspiration. The explanted tissue was allowed to attach to the fibronectin substrate for 12–24 h in a 37˚C incubator with 5% $CO_2$ in a minimum volume of HBSS while avoiding drying of the tissue. Dulbecco's modified Eagle's medium (DMEM; Invitrogen) and F12 (100 μL, mixed 1:1; Invitrogen), supplemented with N2, B27 (both from Invitrogen), and ampicillin (50 μg/mL), was deposited gently at the side of the tissue.

To induce excitotoxicity, explants from 12 animals were supplemented with kainate (Sigma, Saint Louis, MO, cat. #K2389) for 2 h at a concentration of 0.4 mM [9]. Twelve different explants received no treatment and served as controls. Both the treated and control explants were divided into 3 subgroups (n = 4 for each), and the subgroups were cultured for 5, 24, and 72 h, respectively. At each time point, the cultured explants were dissected into the spiral ganglion (SG) and the remaining organ of Corti (OC), and were separately subjected to RNA extraction.

## Extraction of total RNA and cDNA synthesis

Total RNA was purified from the tissue using RNeasy spin-columns (Qiagen) according to the manufacturer's protocol [20]. Specifically, Trizol was used first, followed by chloroform, collection of the aqueous phase, addition of an equal volume of isopropanol, and application of this mixture to RNeasy spin-columns. The RNA quantity and quality was assessed using an Agilent 2100 Bioanalyzer (Agilent Technologies) and RNA Pico Kit (Agilent Technologies); samples with RNA integrity number of at least 7, with 10 being the purest, were used for further analysis. Total RNA that appeared clean and undegraded, based on the Bioanalyzer's electropherograms, was reverse transcribed with Taqman Reverse Transcription Reagents kit (Applied Biosystems). On average, RNA from 3.2 tissue samples (range: 2–4) in each time-specific subgroup met our quality criteria, and was reverse transcribed to cDNA for microarray analysis or qPCR.

## Microarray transcript analysis

Comparative microarray analysis was performed using MouseRef8 v1.1 Expression BeadChip Illumina Arrays, as described previously [21]. The data have been deposited in GEO under accession number GSE130495. Each of these arrays has >24,000 mouse targets based on the NCBI mouse Reference Sequence Database, including 16,287 constitutive exons/islands based on the splice variants in the mouse transcriptome (Molecular Signature Database; MouSDB3) and NCBI LocusLink databases. Illumina arrays provide a detection p-value (0 = max confidence in a gene being detected) for each of the probes. If using 0 as a threshold to call a gene present, 11,991/25,697 (47%) probes are called present in at least one sample in the organ of Corti, and 12,165/25,697 (47%) in the spiral ganglion tissue. Differentially expressed genes were classified according to their gene ontology (http://www.geneontology.org/) [22], using DAVID Bioinformatics online tools (Database for Annotation, Visualization and Integrated Discovery; http://david.abcc.ncifcrf.gov/) [23]. Cellular pathway association was analyzed according to the Kyoto Encyclopedia of Genes and Genomes (KEGG) database (http://www.genome.jp/kegg/) and pathway maps according to BioCarta (http://www.biocarta.com/genes/index.asp).

## Ingenuity pathway analysis

Pathway and global functional analyses were performed using Ingenuity Pathway Analysis (Ingenuity Systems, www.ingenuity.com) on July 20, 2014. A data set containing gene identifiers and corresponding expression values was mapped using the Ingenuity Pathways Knowledge Base. This database identifies biological functions as well as the well-characterized, canonical pathways most significant to the data set. Genes from the data sets associated with biological functions or with a canonical pathway in the database that met the p-value cutoff of 0.005 were used to build the networks as described below. Fisher's exact test was used to calculate a p-value determining the probability that each biological function and/or canonical pathway assigned to this data set was not due to chance alone.

## Network analysis of differentially expressed genes

The accession number and the normalized ratio for each gene whose expression was significantly changed were analyzed using the Ingenuity System Database, including the Ingenuity Knowledge Base and the Global Molecular Network. These databases integrate millions of published findings on biologically meaningful genetic or molecular gene/gene product interactions from all available species, tissue and cell lines, and identify functionally related gene networks. We considered both direct interactions, which require that two molecules make direct physical contact with each other, and indirect interactions, which involve an intermediate. Pathway analysis allows the maximal number of molecules per network to be selected as 35, 70 or 140; we chose 35 to facilitate visual tractability of the networks. The system computes a score for each network according to the fit of the set of the supplied focus genes. The scores indicate the likelihood that the associations forming the network are due to chance alone. A score of >2 indicates a $\geq$ 99% confidence that a network with focus genes was not generated by chance. According to the degree of interconnectedness among the molecules, a higher or lower network score is assigned; the higher the score the more significant the network. To focus on very highly significant networks, we selected only networks with a score of 10 or higher for further analysis. Focus genes were denoted with red symbols if upregulated, and with green symbols if downregulated. Grey and open symbols are intermediate molecules. Symbols representing the functional categories of the molecules are listed in the legend of each figure.

## Validation of the genes of interest

Genes of interest revealed by microarray analyses were validated both at the RNA level, using the on-line Shared Harvard Inner-Ear Laboratory Database (https://shield.hms.harvard.edu/), the Auditory and Vestibular Gene Expression Database (http://goodrich.med.harvard.edu/resources/resources_microarray.htm) databases [24], or RNAscope, and at the protein level using immunofluorescence staining.

## Multiplex fluorescence *in situ* RNA detection

Cochleae of neonatal day 4 mice were fixed in 4% paraformaldehyde (PFA), immersed in 10% sucrose, 20% sucrose, 30% sucrose, embedded in OCT compound, cryosectioned into 12 μm thick sections and stored at -80˚C. In situ detection of *Nlgn2* and *Ntrk3* was performed using commercially available RNA scope Fluorescent Multiplex Detection Reagents from Advanced Cell Diagnostics [ACD], Newark, CA following protocols recommended by the manufacturer. Each set of probes contains a tag that enables visualization of the target transcript in a specific color channel. Specifically, fixed frozen cochlear sections were post-fixed for 40 min with 4%

paraformaldehyde, soaked in PBS twice for 5 min each, soaked for a few seconds in distilled water, then rinsed 3 times in 100% ethanol and air dried. Slides were pretreated to complete protease digestion using Protease III & IV Reagents (cat#322340; ACD) in HybEx oven at 40˚C for 50 min with multiple washes. The probe was applied for 2 h at 40˚C in HybEx oven. After washes with wash buffer, 4 amplification steps followed. Nuclei were visualized using DAPI and specimen coverslipped.

## Immunofluorescence

Inner ears were dissected out of the temporal bone and fixed in 4% paraformaldehyde for 20 min (neonatal) or 2h (adult). Cochleae from 8-weeks old mice were decalcified in 0.12M EDTA for 72 h, embedded in paraffin and sectioned. Paraffin-embedded tissue sections were mounted on silane-coated glass slides, deparaffinized in xylene and rehydrated in ethanol. After antigen heat retrieval (Dako S1700, 30 min at 95˚C), the slides were incubated overnight at room temperature with primary antibodies in PBS and Tween (PBST): mouse anti-ELAV-like protein 4, 1:50 (sc-48421, Santa Cruz) and rabbit anti-synapsin I, 1:50 (Abcam) [25, 26]. Slides were washed and incubated for 1 h at room temperature with the appropriate secondary antibodies at a 1:200 dilution in PBST. After incubation, the slides were washed with PBST and mounted with the VECTASHIELD mounting medium (Vector Labs, CA) at room temperature. Images were obtained by epifluorescence microscopy (Axioskop 2 Mot Axiocam, Zeiss, Germany). Control slides, with the primary antibodies omitted, were processed in parallel. Whole mounts of the organ of Corti were prepared on neonatal day 4, blocked with 5% normal horse serum (NHS) and 0.3% Triton X-100 (TX-100) in PBS for 1 h at room temperature, and immunostained overnight at room temperature with the following primary antibodies diluted in 1% normal horse serum with 0.3% TX: mouse anti-ELAV-like protein 4 at 1:10000 (sc-48421, Santa Cruz), rabbit anti-synapsin I at 1:500 (Abcam, ab64581), rabbit anti-myosin 7A at 1:500 (#25–6790 Proteus Biosciences) or mouse anti-myosin 7A at 1:10 (Developmental Studies Hybridoma Bank, IA, 138–1) to label hair cells, chicken anti-NF-H at 1:2500 (Millipore; #AB5539) to label neurites, mouse (IgG1) anti-CtBP2 (C-terminal Binding Protein) at 1:1000 (#612044, BD Transduction Labs) to label pre-synaptic ribbons and mouse (IgG2a) anti-PSD95 (post-synaptic density 95) at 1:1000 (#75–028, Neuromab) to label post-synaptic patches. After washing in PBS three times, cochlear pieces were incubated in species-appropriate secondary antibodies (Pacific blue-conjugated chicken anti-rabbit, Alexa Fluor 488-conjugated anti-rabbit, Alexa Fluor 488-conjugated anti-mouse (IgG2a), Alexa Fluor 568-conjugated anti-chicken, Alexa Fluor 569-conjugated anti-mouse (IgG1), Alexa Fluor 647-conjugated anti-mouse, Alexa Fluor 647-conjugated anti-rabbit (Life Technologies)). After incubation for 1 ½ h at room temperature, nuclei were labeled with Hoechst 33342 (1: 10 000, Invitrogen), specimen washed thrice in PBS and mounted with VECTASHIELD mounting medium (Vector Labs, CA). Specimens were imaged with a Leica SP8 confocal microscope. As a negative control, primary antibodies were omitted from the staining protocol. This resulted in no specific signal. As a positive control, immunostaining using unrelated primary antibodies gave rise to different specific patterns of positive cells than what we observed in the current study. To quantify auditory nerve fiber bundles (ANF) per inner hair cell (IHC), IHCs stained with Myo7A and ANF bundles stained with NF-H were manually counted per 100 μm. ANF-bundles were counted and assessed approximately 10 μm modiolar to IHCs.

## Statistics

Statistical significance was assessed using a one-way or two-way ANOVA, followed by Bonferroni post hoc multiple comparison test as implemented with the Statistics Toolbox Software

MATLAB, Version 6.1 (R2007b), MathWorks (http://www.mathworks.com/help/toolbox/stats/rn/brasjn_.html).

## Results

### Synaptic regeneration after kainate treatment of cochlear explants

The terminal processes of spiral ganglion neurons on hair cells are lost after kainate exposure due to excitotoxic lesioning of the synapses in the cochlea, but regenerate over the next 72 h [9]. The time course of the model as described in the rat was reproduced in mouse ears (Fig 1). Damage was extensive at 5 h and reinnervation of hair cells by the peripheral processes of the neurons and the formation of PSD-95 and CtBP2-positive synapses was apparent at 24 h. Growth was complete before 72 h, as the processes did not produce further reinnervation at this time point, and the peripheral processes were reduced in number in both the kainate-treated and control samples, suggesting some deleterious effect of the prolonged culture.

The RNA prepared from these samples was divided into the spiral ganglion (SG) and the organ of Corti (OC). RNA was isolated from each element at time points representing damage (5 h), ongoing regeneration (24 h), and completion of regeneration (72 h).

### Internal validity of the microarray data

The heat-map of the samples assayed by microarrays is shown in Fig 2. The spiral ganglion samples were separate from the organ of Corti samples as a coherent group. Moreover, tissue samples obtained at the same time points after kainate treatment clustered together. Clustering based on inter-array Pearson correlation coefficient indicated no batch effects.

### Effects of kainate on gene expression in the explanted tissues

Microarray analysis with a threshold of $p < 0.005$ revealed that a total of 2.5% (588 out of 24,000 transcripts) of the Illumina gene probes were differentially regulated after kainate treatment (Fig 3A). Of note, a large number (n = 282) of gene probes showed significant change in the SG fraction 24 h after kainate treatment, coinciding with the timing of neuronal regeneration; of the 282 transcripts, 171 were up-regulated, and 111 were down-regulated.

Among the 588 gene probes that showed significant changes with kainate treatment, 417 were specific to the SG tissues, 168 were specific to the OC tissues, and 3 changed in both tissues, 2 at 24 h and 1 at 72 h (Fig 3A).

The groups of genes for which the expression changed significantly were subjected to gene ontology analysis, allowing functional annotation using the DAVID Bioinformatics Resources and the KEGG pathways. Major biological processes in the SG tissue ascribed by gene ontology at 5 and 24 h respectively were cell migration/chemotaxis, and regulation of nervous system development/cell communication (Fig 3B).

### Expression profiles of known neuronal regeneration-related genes

We searched for significant changes in genes with known roles in neural regeneration and development at 5 or 24 h. Six showed significant changes ($p < 0.005$) after kainate treatment. These genes are *Nlgn2* (Gene ID: 216856), *Ntrk3* (Gene ID: 18213), *Rgmb* (Gene ID: 68799), *Sema7a* (Gene ID: 20361), *Sox21* (Gene ID: 223227), and *Syt8* (Gene ID: 55925). Except for *Sox21*, which showed significant change in organ of Corti, all of the changes were in spiral ganglion tissues (Fig 4). The expression profiles of these genes are further detailed in Table 1 and show upregulation of *Nlgn2*, *Ntrk3*, *Rgmb*, and *Sema7a* and downregulation of *Syt8*. The 5 genes with significant changes in the spiral ganglion have been reported to have abundant

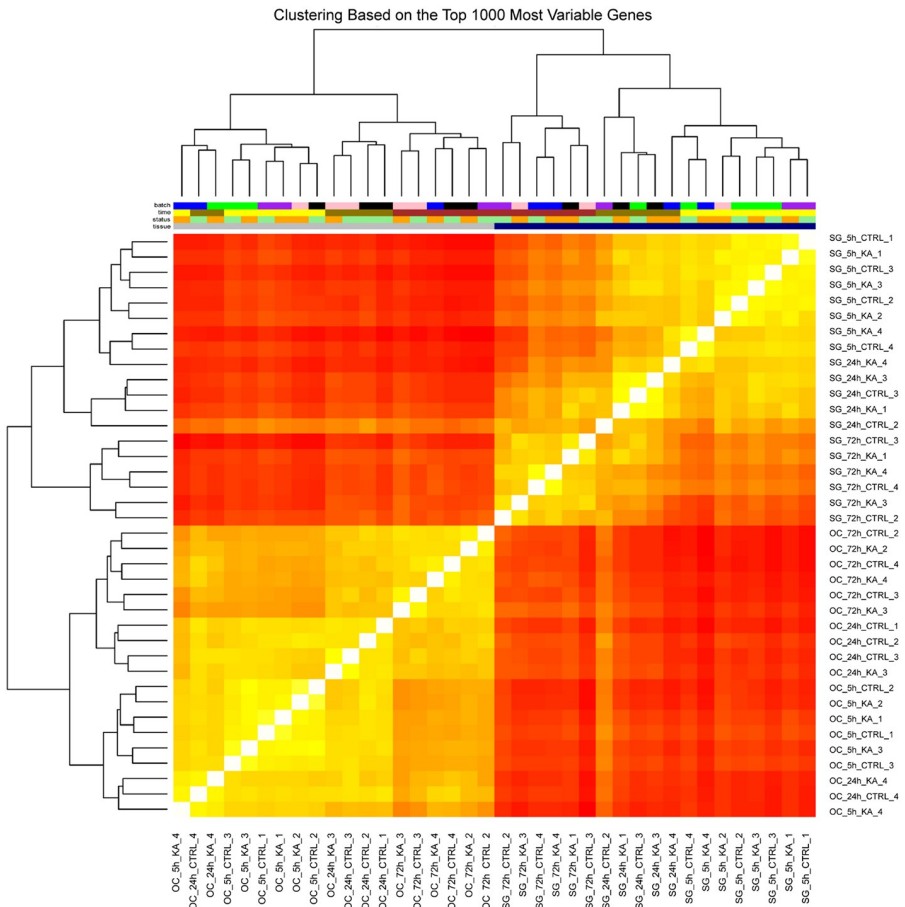

**Fig 2. Clustering of the samples assayed by microarrays.** Unbiased hierarchical clustering of spiral ganglion (SG) and organ of Corti (OC) samples, based on the top 1,000 most variable genes. Each cell represents a comparison between two samples, and its color is proportional to the Pearson correlation across the top 1,000 most variable genes (red: high correlation, yellow: low correlation). Top: samples are color-coded by batch, timepoint, treatment, and tissue. Tissue source was the primary determinant of clustering, followed by partial clustering by timepoint. For better readability of sample names, we have included S1 Table.

RNA expression in the spiral ganglion database from embryonic mice [24]. The one gene that is expressed only in supporting cells of the organ of Corti (*Sox21*) is not found in the spiral ganglion database at P1.

These 5 genes play a role in fiber growth (*Ntrk3*), axonal guidance (*Rgmb* and *Sema7a*), synaptogenesis (*Nlgn2*), or synaptic function (*Syt8*).

## Genes with significant changes at least 2 time points after kainate treatment

We then searched for genes with transcripts with significant changes at $\geq 2$ different time points after kainate treatment, because these genes are active throughout the regeneration process. In contrast to the OC tissues, where no transcripts showed significant changes at $\geq 2$ time points, 8 transcripts revealed significant changes at 2 time points in the SG tissues (Fig 5). Five transcripts showed significant changes at 5 and 24 h, including 1 un-annotated transcript

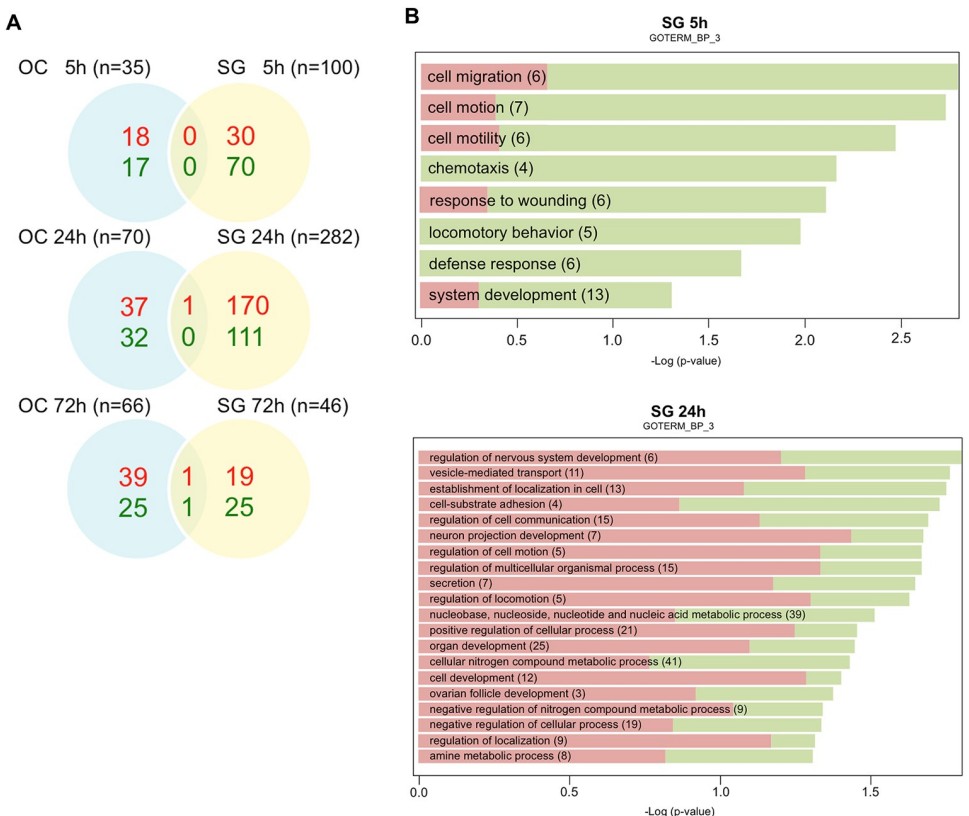

**Fig 3. Gene expression changes in cochlear explants after kainate treatment.** (A) Differentially expressed genes after kainate treatment. mRNA from spiral ganglion neurons versus organ of Corti fractions was analyzed with microarrays at 5, 24, and 72 h after kainate treatment. Venn diagrams represent the numbers of transcripts that were either upregulated (red) or downregulated (green) at a significance of p<0.005 following kainate treatment relative to untreated controls at 5, 24, and 72 hours after treatment. (B) Gene Ontology (GO) analysis representing the major biological processes involved in the SG tissues at 5 and 24 h after kainate treatment. Over-represented GO categories among differentially expressed (DE) genes (in green the proportion of downregulated DE genes; in red the proportion of upregulated DE genes) sorted by -log10 (p value). The value 1.3 on the horizontal axis corresponds to p = 0.05. OC, organ of Corti; SG, spiral ganglion.

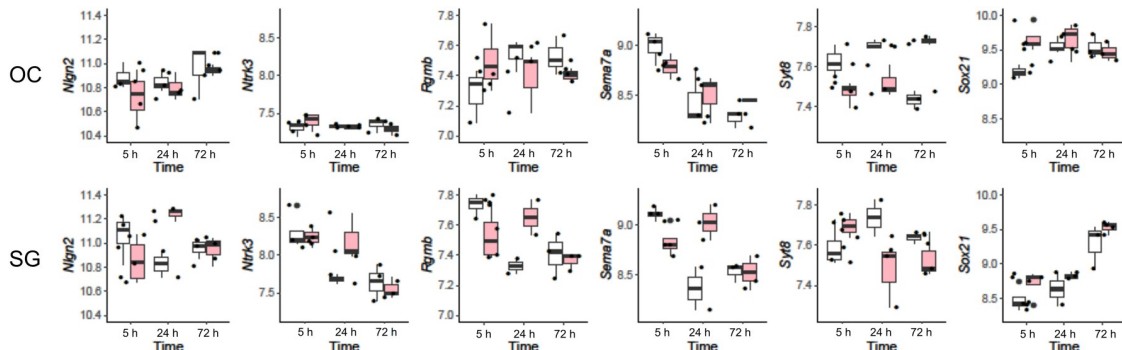

**Fig 4. Expression profiles of 6 neuronal regeneration-related genes with significant changes after kainate treatment.** Expression levels of 6 transcripts (*Nlgn2*, *Ntrk3*, *Rgmb*, *Sema7a*, *Sox21*, and *Syt8*) in KA-treated (red) and control (clear boxplots) tissue samples of organ of Corti (OC, top row) and spiral ganglion (SG, bottom row) harvested at 5, 24 and 72 hours, are represented by boxplot and superimposed scatterplot.

**Table 1. Five neuronal regeneration-related genes with significant changes in the spiral ganglion after kainate treatment.**

| Gene | Protein | Functions in neurons | Fold change vs. control | | |
|---|---|---|---|---|---|
| | | | 5 h | 24h | 72h |
| *Nlgn2* | Neuroligin-2 | Involved in synapse formation, maturation and specification | -1.6 | 2.6* | -1.0 |
| *Ntrk3* | Neurotrophic tyrosine kinase receptor type 3 (alias TrkC) | Promotes spiral ganglion neuronal survival in the inner ear | 1.2 | 3.3* | -0.8 |
| *Rgmb* | Repulsive guidance molecule family member B | Regulates neurite growth and axonal guidance; contributes to the patterning of the developing nervous system. | -1.6 | 2.4* | -0.9 |
| *Sema7a* | Semaphorin7A | Involved in neuron migration and neurite growth | -1.9 | 4.5* | -1.4 |
| *Syt8* | Synaptotagmin 8 | Unclear | 1.3 | -1.8* | -1.3 |

\* p < 0.005 as compared to the controls.

Negative numbers indicate downregulation and positive numbers indicate upregulation of gene expression relative to untreated controls.

A830007L07Rik, and 4 known genes: *Elavl4* (Gene ID: 15572), *Ppp1r1c* (Gene ID: 75276), *Psd2* (Gene ID: 74002), and *Syn1* (Gene ID: 20964). Interestingly, all 4 genes demonstrated a similar temporal expression pattern: a significant decrease in expression at 5 h as compared to the untreated control tissues, followed by a significant increase in expression at 24 h (Table 2). This time course correlates with the neuronal regeneration process. According to the SHIELD

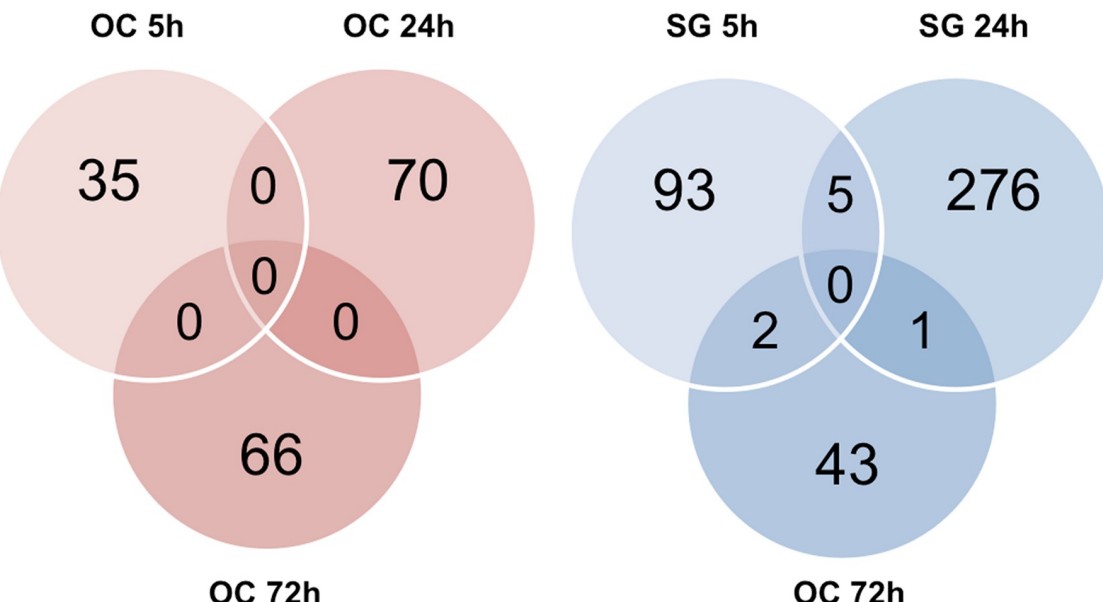

**Fig 5. Venn diagrams representing the numbers of transcripts with significant changes at different time points after kainate treatment.** In contrast to the organ of Corti tissues where no transcript significantly changed at more than one time points (left panel), 8 transcripts significantly changed at two different time points in the spiral ganglion tissues (right panel). OC, organ of Corti; SG, spiral ganglion.

**Table 2. Four genes with significant changes at two different time points after kainate treatment.**

| Gene | Protein | Function in neurons | Fold change vs. control 5 h | 24 h |
|------|---------|---------------------|------|------|
| *Elavl4* | ELAV-like protein 4 | Control neuronal development and functions by regulating RNA metabolism. | -3.4* | 4.2* |
| *Ppp1r1c* | protein phosphatase 1, regulatory subunit 1C | Inhibit neurite growth in primary sensory neurons by maintaining TGF-ß/Smad signaling | -3.4* | 4.0* |
| *Psd2* | pleckstrin and Sec7 domain-containing protein 2 | Regulate axon transport and axon growth in neurons; interact with interferon γ pathway. | -2.4* | 3.1* |
| *Syn1* | synapsin I | Control the transition of synaptic vesicles, regulate neurite outgrowth, and promote neuronal survival. | -2.9* | 4.3* |

* p < 0.005 as compared to the controls.

Negative numbers indicate downregulation and positive numbers indicate upregulation of gene expression relative to untreated controls.

database, all 4 genes showed abundant RNA expression in the spiral ganglion neurons of P0—P6 mice.

To study the molecular mechanisms of *Elavl4*, *Ppp1r1c*, *Psd2*, and *Syn1* in neuronal regeneration, we performed network analysis using the normalized ratios of the 417 SG-specific genes. *Elavl4* and *Syn1* are located in Network 1 (score 29, Fig 6A), further substantiating their

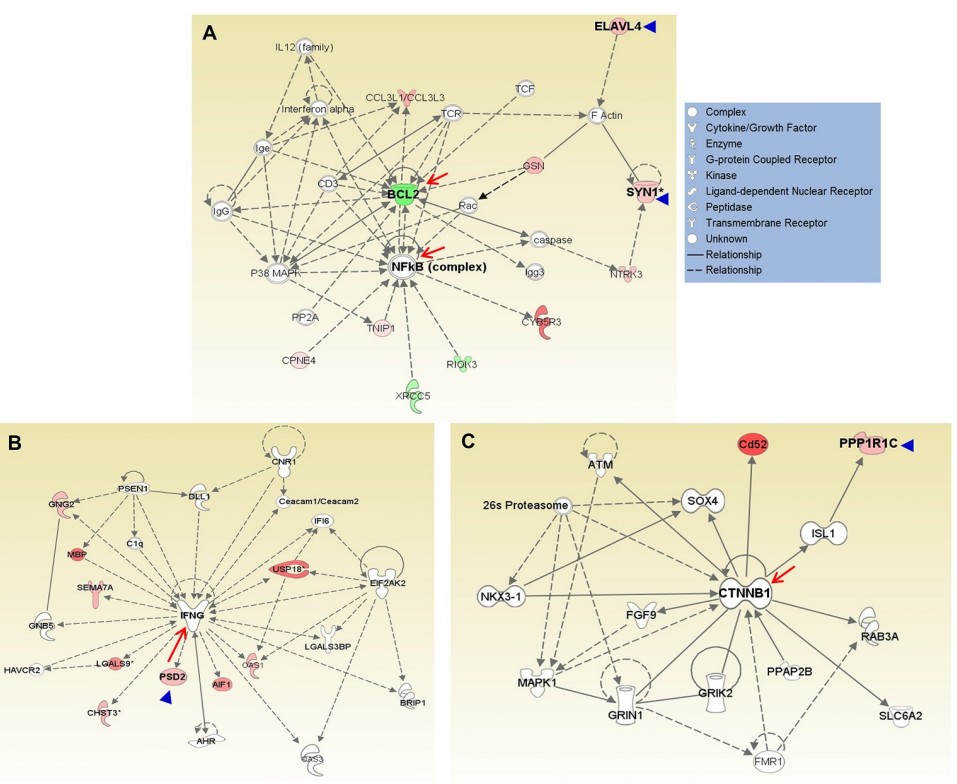

**Fig 6. Networks including Elavl4, Ppp1r1c, Psd2, and Syn1 when analyzing differentially expressed genes in the spiral ganglia after kainate treatment.** A total of 417 transcripts showed significant changes specific to the spiral ganglion tissues after kainate treatment. (A) *Elavl4* and *Syn1* were located in Network 1 (score = 29) with NFkB and BCL2 as the central hubs. (B) *Psd2* was located in a network (score = 19) with interferon γ as the hub. (C) *Ppp1r1c* was located in a network (score = 11) with CTNNB1 as the hub. Arrow heads indicate the 4 genes with consistently significant changes at both 5 h and 24 h after kainate treatment; arrows indicate molecular hubs of the networks.

biological significance in regeneration of spiral ganglion. This network contains 21 focus molecules and has two hubs, NFkB and BCL2, with 14 and 13 connections, respectively. F-actin, GSN, and Rac appear to be points of interaction, connecting ELAVL4 and SYN1 to NFkB and BCL2; SYN1, NTRK3 and caspases also contribute to the connection. The molecular functions of this network are related to cellular assembly, organization, and maintenance, as well as cell signaling.

*Psd2* is found in a network with a score of 19 and 16 focus molecules (Fig 6B). This network shows interferon γ as the hub with 22 connections, including a connection to *Psd2*. *Ppp1r1c* is located in a network with a score of 11 and 11 focus molecules (Fig 6C). The hub of this network is catenin (cadherin-associated protein), ß1, 88kDa (CTNNB1), with 15 connections. ß-catenin is the intracellular mediator of the canonical WNT signaling pathway as well as a regulator of cadherin-based adherens junctions. *Ppp1r1c* is linked to CTNNB1 via ISL-1. The molecular functions of this network are related to cell-to-cell signaling and interaction, cellular function and maintenance, as well as nervous system development and function.

To further investigate these genes in the cochlea, we conducted immunofluorescence staining of cochlear whole-mounts at P4 and sections at 8 weeks. Synapsin I, encoded by *Syn1*, was located at the afferent synapses in neonantal (Fig 7A and 7B) and young adult mice (Fig 7C). ELAV-like protein 4, the protein encoded by *Elavl4*, was strongly expressed in spiral ganglion neurons of neonatal (Fig 7D and 7E) and young adult mice (Fig 7F).

To further validate expression of genes for whose protein products robust commercial antibodies did not exist, we used in situ hybridization. Application of negative-control sense probes for *Nlgn2* and *Ntrk3* to P4 cochlear explants did not result in any specific signal, as illustrated for *Nlgn2* in Fig 8A. In contrast, anti-sense RNA probes allowed identification of individual molecules of *Nlgn2* (Fig 8B) and *Ntrk3* (Fig 8C) within the area of spiral ganglion neurons but not hair cells, which were identified with anti-Myo7a antibody. Data in Fig 8 are representative of experiments from 4 different animals.

## Discussion

### Model system for auditory nerve regeneration

Regeneration of afferent auditory fibers is induced in neonatal cochleae after kainate treatment. We used this system because we were interested in the genetic changes in the de-afferented system, where kainate causes swelling of the type I afferent dendrites that synapse on inner hair cells, without noticeably damage to efferent endings or type II afferent terminals that synapse on outer hair cells. While ~95% of afferent spiral ganglion neurons are of type I and synapse to one inner hair cell, type II afferents contact multiple outer hair cells [27]. As a potent glutamate analogue, kainate has been extensively used to investigate excitotoxicity in neurons. Kainate exerts neuroexcitotoxicity by acting on glutamate receptors, eliciting influx of calcium ions, production of reactive oxygen species, and apparent breakdown of the neural terminals. Active neuronal regeneration occurs after kainate treatment: the swollen afferent dendrites are replaced by regenerating dendrites within 12–72 h.

### Changes in spiral ganglion and organ of Corti

Genes with significant changes in the spiral ganglion tissues by far outnumbered, and were distinct from those in the organ of Corti. This correlated well with the demonstration of kainate-induced damage confined to the auditory nerve [9]. More gene transcripts changed (n = 282) in SG obtained 24 h after kainate treatment than at 5 (n = 100) or 72 h (n = 46), coinciding with the timing of active neuronal regeneration. We utilize *in situ* hybridization (Fig 8) and

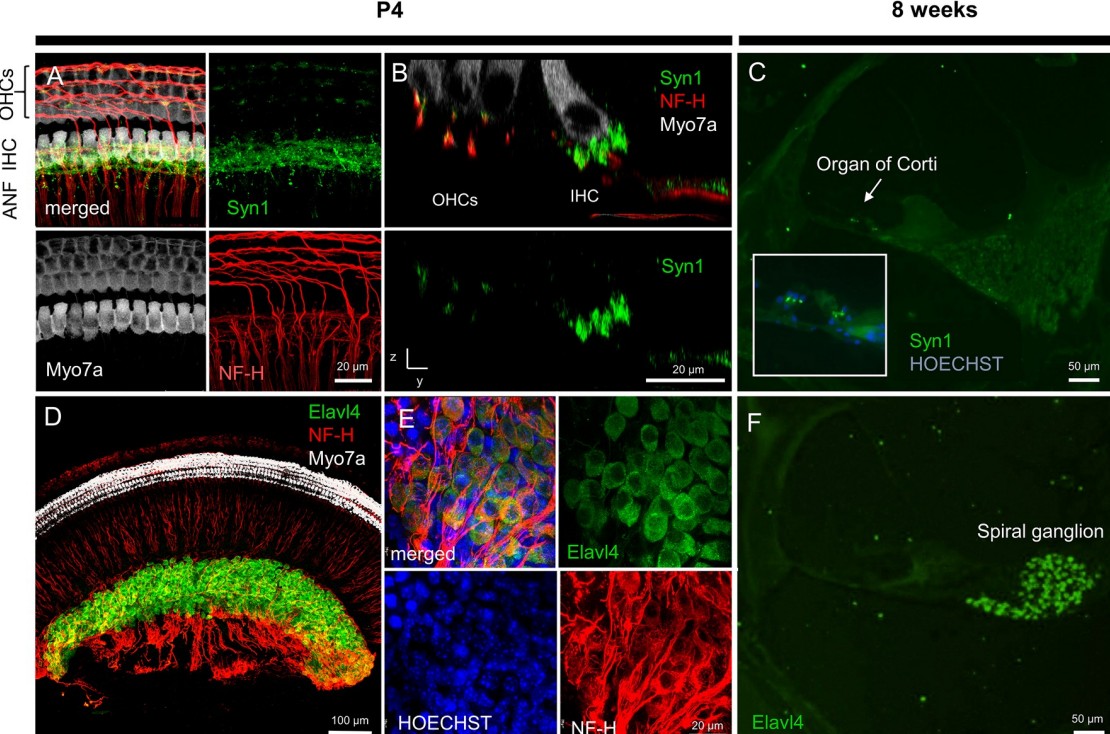

**Fig 7. Immunofluorescent staining of the protein products of *Elavl4* and *Syn1* in the mouse inner ear at P4 and 8 weeks of age.**
(A) Fluorescent immunohistochemistry of cochlear whole mounts at P4 revealed the expression of synapsin I (green) in the basal area of IHCs as well as OHCs (both white, labeled with anti-Myo7a antibody). Red, NF-H-labeled auditory nerve fibers. (B) A virtual cross-section of Fig A reveals synapsin 1 fluorescence localization at the IHC-SGN synapse, not overlapping with NF-H (red) or Myo7a (white) signal, and at the OHC-SGN synapse, partially overlapping with NF-H (red) but not Myo7a (white) signal. (C) At 8 weeks of age, synapsin 1 expression is localized to the organ of Corti at the base of IHCs and OHCs (nuclei stained blue, HOECHST). (D) Low magnification view of a cochlear whole mount shows strong Elavl4 signal in the modiolus. (E) A magnified view of D reveals Elavl4 expression in SGNs stained with NF-H (red). (F) Eight-weeks old cochlea reveals Elavl4 immunofluorescence in spiral ganglion neurons. Scale bars: A, B: 20 μm, C: 50 μm, D: 100 μm, E: 20 μm, F: 50 μm. Representative images from N = 3 four-day-old mice and N = 3 eight-week-old mice.

immunofluorescence (Fig 7) to specify the localization of genes of interest or of proteins these genes encode for, as illustrated in Fig 9.

## Inflammatory/immune response and neuronal regeneration

One of the conclusions from the gene families that were increased was the preponderance of genes in the inflammation/immune response family. The swelling and bursting of neural endings in response to excess glutamate triggers the expression of genes related to inflammation in the cochlea.

Neuronal regeneration in the inner ear after kainate treatment, as shown in the gene ontology analysis, appears to occur in 2 phases: the first phase (within hours after kainate treatment) represents inflammation and involves differential expression of genes related to cell migration and chemotaxis; the second phase (~24 h after kainate treatment) involves neuronal regeneration as reflected in changes in genes of nervous system development and cell communication.

The relationship of inflammation to regeneration has been recognized. One aspect of this relationship is the potentially beneficial effect of inflammatory response genes on nervous system regeneration. Indeed the lack of an inflammatory response can retard regeneration [28], and spinal cord regeneration has been noted in response to macrophage infiltration and

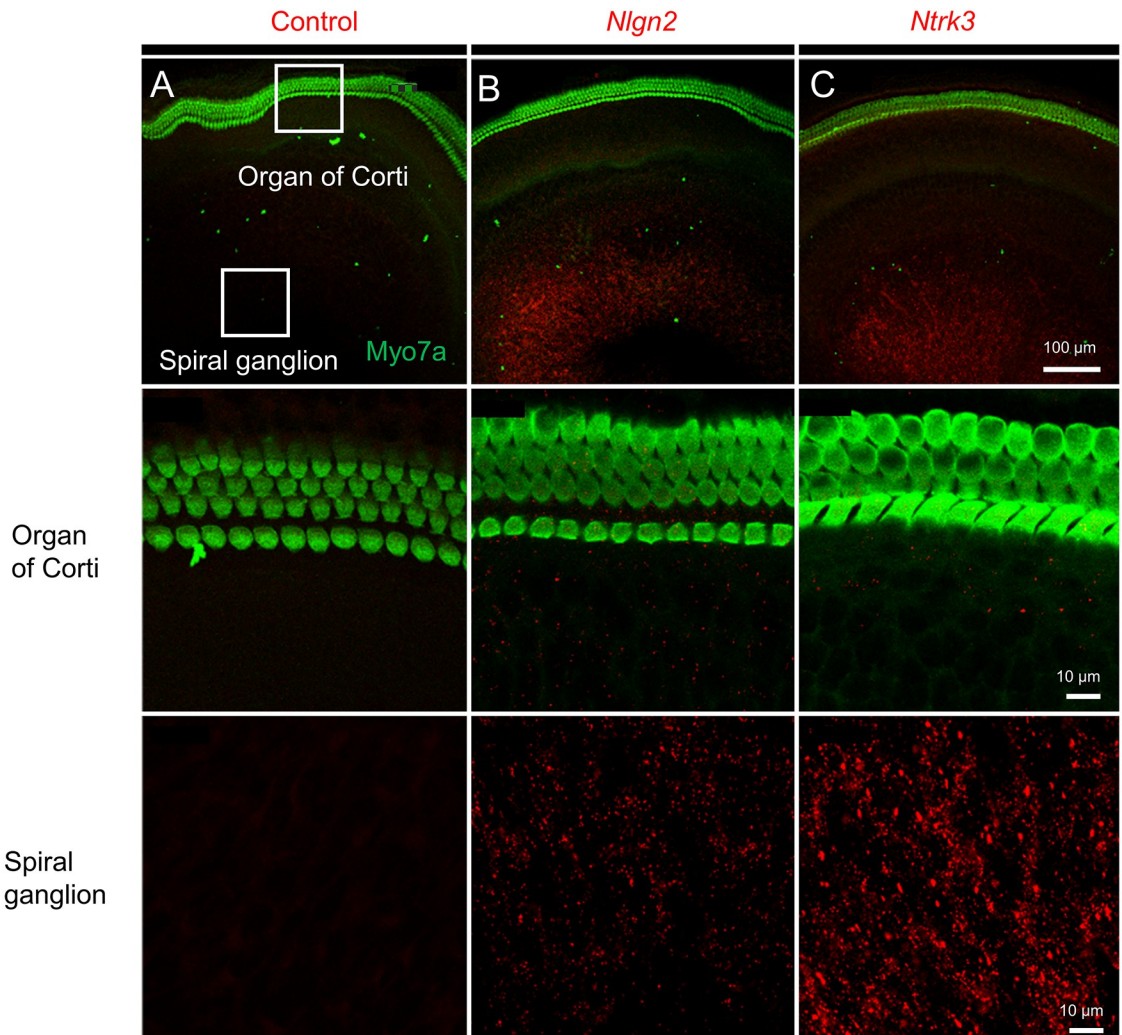

**Fig 8. In situ hybridization assay utilizing RNAscope detects RNA for *Nlgn2* and *Ntrk3* within P4 cochlear whole mounts.** (A) Cochlear whole mounts treated with negative control sense probe for *Nlgn2* show no signal. Hair cells were immunostained with anti-Myo7a antibody (green). (B) *Nlgn2* is expressed in the region of spiral ganglion neurons (red), but not hair cells (green). A magnified view of the SGN region (last row) identifies individual RNA molecules of *Nlgn2* as red dots. (C) *Ntrk3* is expressed in the SGN region (red), but not hair cells (H, green). A magnified view of the SGN region (last row) reveals abundant *Ntrk3* expression. Representative pictures based on 4 different animals.

cytokine release [29] prompting the theory that some of the elements recognized as immune mediators may play an additional role in the regenerative response. An example of that is the complement system, which has been shown to play a role in synaptic pruning [30]. Furthermore, cochlear supporting cells which surround auditory nerve fibers and are capable of trans-differentiating into hair cells express at baseline a pro-inflammatory cytokine, CXCL1 [31].

## Specific genes altered in neurogenesis and synaptogenesis

To clarify the mechanisms underlying the temporal changes in the biological processes after kainate treatment, we further investigated transcripts with significant changes over time. Some of these genes were increased significantly in the spiral ganglion or organ of Corti tissues (Table 1), whereas others were significantly changed at least 2 time points after treatment

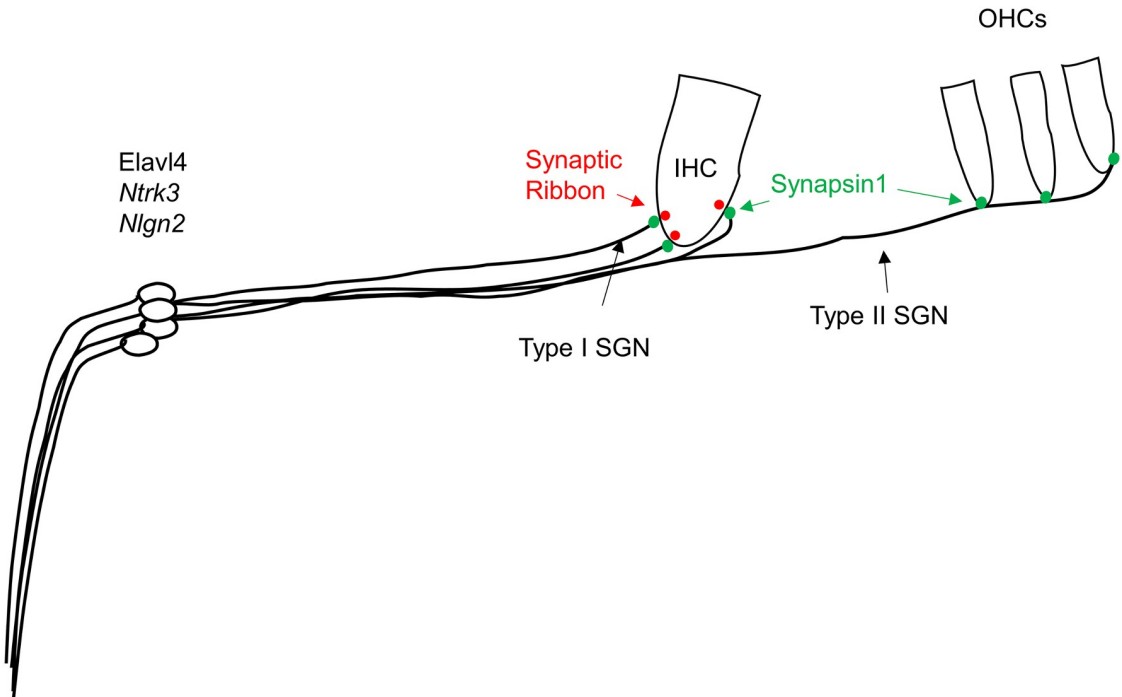

**Fig 9. Schematic illustrating localization of genes of interest.** ELAV-like protein 4, encoded by *Elavl4*, as well *Ntrk3* and *Nlgn2* localize to the spiral ganglion area. Synapsin 1 (green), encoded by *Syn1*, was found to be present at the synapse between spiral ganglion neurons with inner and outer hair cells.

(Table 2). Genes with a known role in neurogenesis, synaptogenesis, neuronal growth, or axon guidance that were on the list of significantly changed genes included: *Sox21*, expressed in cochlear supporting cells and required for their development; *Nlgn2*, a gene required for synaptogenesis but not previously reported in the afferent auditory synapse; *Ntrk3*, the key receptor for cochlear neurotrophin, neurotrophin-3 (NT3) [32]; *Rgmb*, homologue of *Rgma* which has previously been shown to prevent fiber growth and synaptogenesis of afferent spiral ganglion neuronal fibers [7]; *Sema7a*, a guidance factor that has previously been shown to link neural regeneration to inflammation [33]; and *Syt8*, a synaptotagmin associated with calcium-mediated vesicle formation (Table 1). By contrast, downregulated *Syt8* at 24 h after kainate treatment is not easy to understand without invoking post-transcriptional and post-translational modifications. Genes with significant changes at 2 different time points after kainate treatment included: *Elavl4*, an RNA-binding protein; *Ppp1r1c*, part of a phosphatase expressed in sensory neurons; *Psd2*, a guanine nucleotide exchange factor found in the brain; and *Syn1*, a protein involved in synaptic vesicle release (Table 2). Based on these lines of evidence, it is likely that upregulated expression of these proteins may contribute to neuronal regeneration at 24 h.

The expression and functions of *Elavl4*, *Ppp1r1c*, and *Psd2*, which showed significant changes at both 5 and 24 h after kainate treatment (Table 2), have not been previously explored in the inner ear. Furthermore, while *Syn1* was previously described in the ear by western blot [34], its precise intracellular localization has not been reported. All demonstrated a significant decrease in expression at 5 h, followed by a significant increase in expression at 24 h—this expression pattern correlates closely with the neuronal regeneration process observed histologically. Immunofluorescent studies confirmed the expression of *Elavl4* and *Syn1* expression in the spiral ganglion and/or organ of Corti of P4 neonatal and 8-week old mice. The hubs of the

network containing *Elavl4* and *Syn1*, BCL2 and NFkB (Fig 6A), are known to play critical roles in neuronal survival [35]. In addition, the hub of the network containing *Ppp1r1c* is ß-catenin, CTNNB1, the intracellular mediator of the WNT pathway, which plays a critical role in cochlear progenitors as well as neurogenesis [36, 37] and synapse formation [38]. Protein phosphatase 1 regulatory subunit 1C is highly expressed in the primary sensory neurons of rat dorsal root ganglion and inhibits neurite growth in primary sensory neurons by maintaining TGF-ß/Smad signaling [39]. *Elavl4* encodes ELAV-like protein 4 (alias HuD), an RNA-binding protein responsible for pre-mRNA processing, mRNA stability, and translation. Through its ability to regulate mRNA metabolism of diverse groups of functionally similar genes, ELAV-like protein 4 mediates neuronal development [40]. ELAV-like protein 4, by its linkage to BCL2 and NFkB, is likely to regulate interacting molecules such as F-actin, GSN (gelsolin), and/or Rac. The connection to BCL2 and NFkB also suggests an anti-apoptotic effect [41–43]. Its direct interaction with *Bdnf* mRNA may facilitate location- and activity-specific neuronal BDNF synthesis [25]. *Syn1* (synapsin 1) is abundant in mature mammalian neurons [44]. In addition to controlling the transition of synaptic vesicles from the reserve pool to the readily releasable pool through a phosphorylation-dependent regulation of vesicle-actin interactions [45–47], synapsin I promotes neurite outgrowth and neuronal survival when released via glia-derived exosomes [48].

The biological roles of these upregulated genes after kainate treatment can be generally classified into 5 categories, including neurogenesis (*Elavl4* and *Sox21*), neural outgrowth (*Ntrk3* and *Ppp1r1c*), axonal guidance (*Rgmb* and *Sema7a*), synaptogenesis (*Nlgn2* and *Psd2*), and synaptic vesicular function (*Syt8* and *Syn1*).

**Neurogenesis.** As shown in Fig 6A, *Elav4* is linked to BCL2 and NFkB, and could play a regulatory role in neuronal regeneration by regulating apoptosis in the inner ear, or regulating interacting molecules such as F-actin, GSN (gelsolin), and/or Rac.

*Sox21* is an exception in that it is the only neurogenesis gene that we found upregulated in the organ of Corti [49, 50]. *Sox21* is expressed in supporting cells at P1 [51]. The organ of Corti at P1 undergoes regeneration of hair cells and this can occur in response to damage [52]. *Sox21* is also the only RNA that comes up at 5 h—this early response in the newborn animal could be an attempt to initiate neurogenesis in response to damage.

**Neural regeneration and survival.** *Ntrk3* (encoding neurotrophic tyrosine kinase receptor type 3, alias TrkC) is the receptor for NT-3, a key factor for neural regeneration. NT-3/TrkC has been shown to promote spiral ganglion neuronal survival in the inner ear [53]. BDNF and NT3 increase regeneration—this suggests that the response is specific to regeneration [32, 54]. NT-3/TrkC has been shown to promote spiral ganglion neuronal survival in the inner ear [53]. We did not find an increase in NT-3, but we did find that the receptor increased its expression. This could play a role in the survival of the damaged neurons and in their response to the gradient of neurotrophins.

*PPP1r1c* could inhibit neurite growth in primary sensory neurons by maintaining TGF-ß/Smad signaling [39].

**Axonal guidance.** *Rgmb* (encoding repulsive guidance molecule family member B) is an axonal growth modulator. Binding of repulsive guidance molecules (RGMs) to neogenin can regulate neurite growth and axonal guidance [55, 56]. Upregulated expression of RGMb has been reported in response to nervous system injury [57]. Its upregulation promotes neurite outgrowth in the sensory dorsal root ganglion and could contribute to BMP signaling in the repair process [58]. This reproduction of the embryonic pattern of inhibition may prevent regrowth of neurons. Our previous work has suggested [7] that inhibitory guidance molecules may prevent the growth of spiral ganglion neurons to hair cells.

*Sema7a* (encoding semaphorin7A) is one of the semaphorins involved in inflammation—it is a link between inflammation and neural regeneration based on cornea [33, 59]. It is upregulated by TGF and BMP [60] and so could be related to the upregulation of RGMb. Semaphorin7A is expressed in many different organ systems, and has been implicated in a wide variety of biological processes, including bone and immune cell regulation, neuron migration, and neurite growth [59].

**Synaptogenesis.** *Nlgn2* (encoding neuroligin-2), is expressed in the membranes of post-synaptic cells and plays a role in synaptogenesis. Neuroligin family members are components of new synapses and are important in cognition for generating new synapses [61]. Neuroligins bind to their presynaptic partners, neurexins, contributing to synapse formation, maturation, and specification [62].

*Psd2* (Pleckstrin and Sec7 domain-containing protein 2, also known as EFA6C) belongs to the EFA6 family of guanine nucleotide exchange factors. *In situ* hybridization of mouse brain sections revealed *Psd2* RNA predominantly in mature Purkinje cells of the cerebellum and the epithelial cells of the choroid plexus [63]. EFA6 proteins can activate ARF6, which is a component of the post-synaptic complex of glutamate synapses. Regulation of its level and expression is associated with synaptogenesis and in directing axon transport and growth [64]. PSD2 might also exert its effects on the spiral ganglion via the interferon γ pathway (Fig 7), as interferon γ can have a neuroprotective function by inducing neurotrophic growth factor production in activated astrocytes [65].

**Synaptic vesicular function.** *Syt8* (encoding synaptotagmin 8) is a member of the calcium sensing proteins involved in vesicular transport and is expressed in neurons and endocrine cells [66], yet its exact function in the synapses remains unclear. It plays a role in calcium-induced membrane fusion during the acrosome reaction. *Syt8* is a member of the calcium sensing proteins involved in vesicular transport and is expressed in neurons and endocrine cells [66], yet its exact function in the synapses remains unclear.

*Syn1* (encoding synapsin 1) is abundant in mature mammalian neurons [44]. Although we have used synapsin 1 as a marker of synapses *in vitro* [67], its localization and function in the inner ear remain unclear. The surge in synapsin 1 expression at 24 h may contribute to synaptic regeneration.

## Conclusion

In conclusion, we demonstrated that the effects of kainate are tissue-specific in the inner ear, resulting in a two-phase response in the spiral ganglion: an acute response consistent with inflammation, followed by an upregulation of neural regeneration genes, and that neuroinflammation may contribute to neurogenesis. Prominent genes upregulated at several time points included genes with roles in neurogenesis (*Elavl4* and *Sox21*), neural outgrowth (*Ntrk3* and *Ppp1r1c*), axonal guidance (*Rgmb* and *Sema7a*), synaptogenesis (*Nlgn2* and *Psd2*), and synaptic vesicular function (*Syt8* and *Syn1*). Our results provide insight into candidates that could be targeted to enhance regeneration in diseased ears in the future and motivate a future study based on single-cell RNAseq to identify specific cell types that undergo changes in gene expression.

## Supporting information

**S1 Table.**
(DOCX)

## Acknowledgments

The authors thank Fuying Gao for assistance with data analysis.

## Author Contributions

**Conceptualization:** Albert S. B. Edge, Konstantina M. Stankovic.

**Data curation:** Giovanni Coppola, Albert S. B. Edge, Konstantina M. Stankovic.

**Formal analysis:** Hsiao-Chun Lin, Albert S. B. Edge, Konstantina M. Stankovic.

**Funding acquisition:** Albert S. B. Edge, Konstantina M. Stankovic.

**Investigation:** Chen-Chi Wu, Aurore Brugeaud, Richard Seist, Hsiao-Chun Lin, Wei-Hsi Yeh, Marco Petrillo, Giovanni Coppola, Albert S. B. Edge, Konstantina M. Stankovic.

**Methodology:** Chen-Chi Wu, Aurore Brugeaud, Hsiao-Chun Lin, Marco Petrillo, Giovanni Coppola, Albert S. B. Edge, Konstantina M. Stankovic.

**Resources:** Konstantina M. Stankovic.

**Supervision:** Albert S. B. Edge, Konstantina M. Stankovic.

**Validation:** Chen-Chi Wu.

**Visualization:** Chen-Chi Wu, Aurore Brugeaud, Richard Seist, Hsiao-Chun Lin, Wei-Hsi Yeh.

**Writing – original draft:** Chen-Chi Wu, Aurore Brugeaud, Albert S. B. Edge, Konstantina M. Stankovic.

**Writing – review & editing:** Richard Seist, Albert S. B. Edge, Konstantina M. Stankovic.

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
