## [Decision Letter · Decision Letter 0]

18 May 2020

PONE-D-20-10023

Altered Expression of Genes Regulating Inflammation and Synaptogenesis during Regrowth of Afferent Neurons to Cochlear Hair Cells

PLOS ONE

Dear Stankovic,

Thank you for submitting your manuscript to PLOS ONE. After careful consideration, we feel that it has merit but does not fully meet PLOS ONE’s publication criteria as it currently stands. Therefore, we invite you to submit a revised version of the manuscript that addresses the points raised during the review process. 

Both reviewers felt that your manuscript presented some interesting and valuable new data on changes in gene expression in the Organ of Corti and spiral ganglion following glutamate-induced excitotoxicity.  They have also carefully outlined some suggested changes to clarify some details of your manuscript and broaden its interest.   In your revised manuscript, please address each comment specifically.      

We would appreciate receiving your revised manuscript by Jul 02 2020 11:59PM. To enhance the reproducibility of your results, we recommend that if applicable you deposit your laboratory protocols in protocols.io, where a protocol can be assigned its own identifier (DOI) such that it can be cited independently in the future. For instructions see: http://journals.plos.org/plosone/s/submission-guidelines#loc-laboratory-protocols

We look forward to receiving your revised manuscript.

Kind regards,

Clifford R. Hume, MD PHD

Academic Editor

PLOS ONE

2. To comply with PLOS ONE submissions requirements, please provide methods of sacrifice in the Methods section of your manuscript.

'A.E. is a cofounder of and scientific advisor to Decibel Therapeutics and Audion Therapeutics.'

Reviewers' comments:

Reviewer's Responses to Questions

**Comments to the Author**

1. Is the manuscript technically sound, and do the data support the conclusions?

Reviewer #1: Partly

Reviewer #2: Yes

2. Has the statistical analysis been performed appropriately and rigorously? 

Reviewer #1: Yes

Reviewer #2: Yes

3. Have the authors made all data underlying the findings in their manuscript fully available?

Reviewer #1: Yes

Reviewer #2: Yes

4. Is the manuscript presented in an intelligible fashion and written in standard English?

Reviewer #1: Yes

Reviewer #2: Yes

5. Review Comments to the Author

Reviewer #1: This is a concise, yet abbreviated study to examine changes in gene expression to injury and repair in the cochlea. The interesting gene expression results are presented succinctly and will be of interest to the research community. However, quantitation of the injury model and in situ hybridization experiments was missing and evaluation of immunolabeling patterns was incomplete.

Comments:

Lines 56-57. The authors should clarify whether the entire nerve is unable to regenerate or only a sub-class of them.

Line 63. Has RGMa been defined?

Line 86. The authors should state the rationale for using C57BL/6 mice that develop early onset hearing loss rather than a normal hearing mouse, such as CBA.

Lines 197-198. Evidence should be given for the specificity of the anti-Elav14 and anti-synapsin I antibodies since they are utilized in the study to identify the locations of the proteins in the cochlea.

Figure 1. While this figure appears to reiterate previous findings, quantitative analysis of the results is required to show consistency.

Figure 7a-c. The exact location of anti-synapsin I is not clear in the figure, nor is the staining pattern explained in results.

Figure 7d. The apparent tonotopic labeling of the anti-Elab14 antibody should be addressed.

Figure 8B. The signal was not clear in the figure.

Figure 8. Quantitative analysis of the 4 iterations is missing.

Reviewer #2: This paper describes changes in gene expression that occur in two tissues of the neonatal cochlea – the auditory membranous labyrinth (including organ of Corti) and the spiral ganglion - following glutamate-induced excitotoxicity. The authors generated a set of transcriptomics data that will be helpful to other investigators who are interested in exploring the possible regulators of peripheral nerve fiber regeneration after this type of injury. Additionally, the paper serves as a baseline for a future paper in which the authors will explore changes that occur in the mature cochlea with a similar stimulus, after which nerves are not regenerated.

One limitation of this paper is that it is highly focused on only a handful of genes, some of whose expression is already characterized in the cochlea. Plus, the authors did not test the function of any of the genes it describes. The paper is also rambling and unconcise in some places. Below, I provide comments and suggestions that address these aspects of the paper.

General comments

For the general reader, it’s important to describe how mature the cochlear innervation is at the time of explantation. Have all neurites reached their mature positions? Are synapses mature?

Type I and II afferents are referred to for the first time in the Discussion; they should either be described in the Introduction or defined in the Discussion when mentioned.

The terms “link”, “play a role”, “influence”, and “contribute to” are used repeatedly by the authors to describe a connection between a gene/protein and a cellular function. These terms are vague and not helpful for the reader. We would like to know how these genes control the different cellular processes in which they are implicated.

It’s important for the authors to be clear that they are exploring genes that might regulate neurite regeneration not neuron regeneration (neurogenesis). This is confusing at times throughout the paper.

Specific comments

Lines 25-27. The first sentence tripped me up. Spiral ganglion neurons (SGNs) are the only connection between auditory hair cells and the brain. When considering all hair cells, they share equal significance with Scarpa’s ganglion. Some clarifications (immediate focus on auditory system, etc.) would improve the sentence’s accuracy.

For the second sentence, the authors should describe the type of damage they refer to and whether it is inflicted upon SGNs and/or hair cells.

Lines 46-49. References to support claims made in the first 3-4 sentences are lacking.

Line 56. It would be helpful to add “in maturity” to end of sentence “…..terminals showed no evidence for synaptic recovery”.

Line 66: The authors should define kainite when first introduced.

Line 71: This is a transition back to the current study, and this is not clear as written. I suggest you add “In the model we used in this study..”.

Line 72. It is not clear what the authors mean by “reinnervation is approximately 50%”? Only 50% of fibers return? Only 50% of synapses return? Are there any clues as to why only 50%?

Lines 74-76. These two sentences starting at “Comparison to…” seem out of place and would be more helpful at the end of page 3 (i.e., around lines 68-70).

Line 79: “Elucidates” seems like too strong a word – perhaps “explores” or “begins to characterize”?

Line 106. What fraction of the tissue collected and analyzed comprises the OC, which is defined as the cell population sitting on the basilar membrane? It seems disingenuous to simply refer to this tissue as OC because other cells and tissues are harvested. It would also be helpful to the reader to describe the cell types of the SGN that are isolated (glia, perineural connective tissue, endothelial cells, etc).

Lines 125-129: Is it possible to estimate how many genes, or what % of the genome, are represented?

Line 148-149. What is the cutoff they used for “expressed” for a given gene?

Line 208. Is 8) a typo?

Line 234-36. In this first sentence, the authors should specify they are discussing “peripheral” terminal processes in, I presume, the rat.

Line 281-300 and elsewhere: The molecular terminology is confusing. I recommend spelling out the full gene/protein then providing the abbreviation for every gene when it is first introduced. It is helpful to have both protein and DNA names when comparing to data. Also, this same section is quite long, with some of the content more suited to the Discussion.

Line 284: References are lacking for RGM’s function in nerve regeneration or development. Is there evidence that RGMs inhibit neurite regeneration in other tissues? If so, this should be described.

Line 290. Sema7a is “a link” between inflammation and regeneration, not “the link”.

Lines 324-355. This is a long and complicated section - more so than it needs to be. A few things that might help make it easier to navigate: break it up into smaller paragraphs; describe the full gene name, the family of proteins it encodes, and known function of all genes found in the network before describing their relationships in the networks; move discussion elements (hypotheses about functional significance of relationships in the network, etc.) to Discussion.

Line 359 – missing figure number.

Line 372 – “can be achieved” seems like it is something the experimenter is accomplishing, while the mouse is actually doing so. How about “is induced”?

Line 386 - What is the significance of this temporal difference? What type of genes are differentially expressed in each group, and what might that mean?

Line 391. How do the authors know that the swelling and bursting of the neural endings are the triggers for changes in gene expression? Perhaps it was other cellular changes that occurred in response to kainate treatment? Do they have any ideas about which cell types underwent most gene changes - neurons, glia?

Line 397 – The statement that “the first phase after kainate treatment involves cell migration and chemotaxis and the second involves neuronal regeneration” is simplistic. E.g., it is most likely that no cells in this system were “migrating”. The authors probably mean to say that genes associated with these cellular processes were upregulated, but this does not indicate that these processes are occurring. Indeed, the functions of many genes in a given context are unknown and need to be tested.

Why do the authors surmise that the increase in inflammatory genes is related to regeneration in this study rather than injury? Also, although they discuss the positive effects of inflammation upon regeneration, there are also many papers describing the adverse influences of inflammation on cellular or nerve regeneration. These should be discussed too.

Line 419. The authors should provide references for Ntrk3.

Line 422: What does “Link neurogenesis to inflammation” mean? Have Sema7a or Syt8 been examined in the cochlea? This sentence starting with “By contrast…” doesnt make any sense here.

Line 426. What is “part of a phoshatase” mean?

Generally, the Discussion is long, unfocused, and redundant (e.g., a given gene is discussed in several places). It would be helpful if the authors were to condense this section into 2-3 main sections that succinctly interpret the findings.

Also, some interpretations of the findings are simplistic here. Even though the function of specific genes is implicated in ontology analysis or by past papers, it does not mean that these genes are driving these processes in this context. For instance, the idea that Sox21 upregulation in the P5 OC reflects new neurogenesis is illogical. Nerve regeneration in the SGN has several cellular phases - damage, survival, neurite outgrowth, pruning, synaptogenesis, etc. - none of which include neurogenesis.

Line 458. This section says two apparently contradictory things: the change in Ntrk3 is specific to regeneration but also may promote neuronal survival. These are two different processes.

Figure 2: I suspect the resolution of the text in this figure will be too poor in the final printed version.

Figure 3: It is unclear what the significance of the pink and green regions are in B.

Figure 4. Graphs are very hard to understand. What is represented by each dot? What are the red arrows showing? I cannot tell which genes are significantly different between control and treated. It would be helpful if the authors added X axis labels for SGN (5h, 24h, 72h) and OC (5h, 24h, 72h) and arrange data as a column or dot graph with error bars, adding lines with asterisks to indicate which groups have significant difference relative to control.

Re. Figure 5. The authors do not describe the rationale for identifying genes that overlap in 2 or more groups. Why did they make this choice, and what is the potential functional significance of genes that are altered at both timepoints?

6. PLOS authors have the option to publish the peer review history of their article (what does this mean?). If published, this will include your full peer review and any attached files.

Reviewer #1: No

Reviewer #2: No

---

## [Author Response · Author response to Decision Letter 0]

10 Aug 2020

We thank both reviewers for their time and thoughtful comments. We have addressed comments separately.

Reviewer #1: This is a concise, yet abbreviated study to examine changes in gene expression to injury and repair in the cochlea. The interesting gene expression results are presented succinctly and will be of interest to the research community. However, quantitation of the injury model and in situ hybridization experiments was missing and evaluation of immunolabeling patterns was incomplete.

Comments:

Lines 56-57. The authors should clarify whether the entire nerve is unable to regenerate or only a sub-class of them.

 The revised manuscript clarifies that we are referring to afferent type I SGNs. After synaptopathic noise exposure causing a temporary threshold shift, there is no spontaneous synaptic regeneration in mice (Kujawa & Liberman, 2009). Some nerve regeneration can be seen in chinchillas after permanent damage to the entire organ of Corti (Lawner, 1997).

Kujawa SG, Liberman MC. Adding insult to injury: cochlear nerve degeneration after "temporary" noise-induced hearing loss. J Neurosci. 2009;29(45):14077-14085.

Lawner BE, Harding GW, Bohne BA. Time course of nerve-fiber regeneration in the noise-damaged mammalian cochlea. Int J Dev Neurosci. 1997;15(4-5):601-617.

Line 63. Has RGMa been defined?

The revised manuscript defines RGM as repulsive guidance molecule. We have also corrected the typo specifying that we found RGMb (not RGMa).

Line 86. The authors should state the rationale for using C57BL/6 mice that develop early onset hearing loss rather than a normal hearing mouse, such as CBA.

Since we are studying spontaneous neurite regeneration that occurs in the neonatal cochlea before the onset of hearing, C57BL/6 strain is as relevant as the CBA strain. We used C57BL/6 mice because they generate larger litters than CBA mice, many transgenic lines are generated in C57BL/6 background allowing easy translation of our results to other mouse lines, and we did not study mature or aging mice.

Lines 197-198. Evidence should be given for the specificity of the anti-Elavl4 and anti-synapsin I antibodies since they are utilized in the study to identify the locations of the proteins in the cochlea.

The revised manuscript specifies that negative controls in which primary antibodies were omitted gave no specific signal and positive controls utilizing primary antibodies targeting different epitopes gave distinctly different patterns of immunostaining. In addition, the revised manuscript cites references that established specificity of the antibodies we utilized:

Elalv-4: Vanevski et al.,. HuD interacts with Bdnf mRNA and is essential for activity-induced Bdnf synthesis in dendrites. PLoS One. 2015

Synapsin-1: Boesmans et al.,. Structurally defined signaling in neuro‐glia units in the enteric nervous system. Glia. 2019

Figure 1. While this figure appears to reiterate previous findings, quantitative analysis of the results is required to show consistency.

The revised manuscript includes a new panel 1D that summarizes quantification of ANF bundles per IHC at the 3 times points we studied.

Figure 7a-c. The exact location of anti-synapsin I is not clear in the figure, nor is the staining pattern explained in results.

These panels illustrate that Syn1 is expressed at the post-synaptic density of NF-H-expressing SGNs, juxtaposed to the basolateral surface of Myo7a-expressing IHCs, and to a lesser degree OHCs. A new Fig.9 now schematizes this finding as well as localization of other molecules we studied (Elavl4, Ntrk3, Nlgn2).

Figure 7d. The apparent tonotopic labeling of the anti-Elab14 antibody should be addressed.

We did not observed tonotopic labelling. While the picture presented in the originally submitted manuscript showed slightly higher staining intensity on the far right of a part of a cochlear half turn, this was a staining artifact due to differences in tissue height. We reexamined all of our slides spanning the length of the cochlea and did not see a substantial tonotopic gradient. We have therefore replaced the original image with a more representative one.

Figure 8B. The signal was not clear in the figure.

The figure shows signal that is weak but specific in SGNs. These RNAscope results are consistent with low abundance of Nlgn2 transcripts in our data set and published data sets by others, as summarized in gEAR (https://umgear.org/) and SHIELD data bases (https://shield.hms.harvard.edu/).

Figure 8. Quantitative analysis of the 4 iterations is missing.

The RNAs of interest (red) are present in SGNs, not in hair cells immunostained for Myo7a (in green). To perform quantitative analysis per cell, SGNs would also need to be immunostained. However, it is technically very challenging to perform double immunostaining and in situ hybridization on the same tissue sections, and results may not be as robust as what we have shown with combined single immunostaining and in situ hybridization. Since the objective of this figure is to validate expression of the Nlgn2 gene which had not been previously studied in the ear, and NTrk3 expression has been reported and quantified in other studies as reviewed by Green et al. (2012), we believe that this figure achieves our objective.

Green SH, Bailey E, Wang Q, Davis RL. The Trk A, B, C’s of Neurotrophins in the Cochlea. Anat Rec. 2012;295(11):1877-1895. doi:10.1002/ar.22587

Reviewer #2: This paper describes changes in gene expression that occur in two tissues of the neonatal cochlea – the auditory membranous labyrinth (including organ of Corti) and the spiral ganglion - following glutamate-induced excitotoxicity. The authors generated a set of transcriptomics data that will be helpful to other investigators who are interested in exploring the possible regulators of peripheral nerve fiber regeneration after this type of injury. Additionally, the paper serves as a baseline for a future paper in which the authors will explore changes that occur in the mature cochlea with a similar stimulus, after which nerves are not regenerated.

One limitation of this paper is that it is highly focused on only a handful of genes, some of whose expression is already characterized in the cochlea. Plus, the authors did not test the function of any of the genes it describes. The paper is also rambling and unconcise in some places. Below, I provide comments and suggestions that address these aspects of the paper.

General comments

For the general reader, it’s important to describe how mature the cochlear innervation is at the time of explantation. Have all neurites reached their mature positions? Are synapses mature?

We have added the following sentence to the revised manuscript: “Mouse neonatal cochleae on postnatal day 4, the age of dissection, are at a late developmental stage at which hair cell innervation is refined with neurite retraction and synapse pruning, before the onset of hearing around P10 (Kros, 1998; Huang, 2007).“ (Line 77).

Kros CJ, Ruppersberg JP, Rüsch A. Expression of a potassium current in inner hair cells during development of hearing in mice. Nature. 1998;394(6690):281-284. doi:10.1038/28401

Huang L-C, Thorne PR, Housley GD, Montgomery JM. Spatiotemporal definition of neurite outgrowth, refinement and retraction in the developing mouse cochlea. Development. 2007;134(16):2925-2933. doi:10.1242/dev.001925

Type I and II afferents are referred to for the first time in the Discussion; they should either be described in the Introduction or defined in the Discussion when mentioned.

We have clarified the distinction between type I and type II afferents in the discussion (line 346).

The terms “link”, “play a role”, “influence”, and “contribute to” are used repeatedly by the authors to describe a connection between a gene/protein and a cellular function. These terms are vague and not helpful for the reader. We would like to know how these genes control the different cellular processes in which they are implicated.

These terms are used to describe our inferences as to potential significance of observed gene expression changes. Functional studies would require to establish the role these genes play in cochlear function. However, these studies are beyond the scope of the current paper.

It’s important for the authors to be clear that they are exploring genes that might regulate neurite regeneration not neuron regeneration (neurogenesis). This is confusing at times throughout the paper.

We have clarified this throughout the revised manuscript.

Specific comments

Lines 25-27. The first sentence tripped me up. Spiral ganglion neurons (SGNs) are the only connection between auditory hair cells and the brain. When considering all hair cells, they share equal significance with Scarpa’s ganglion. Some clarifications (immediate focus on auditory system, etc.) would improve the sentence’s accuracy.

For the second sentence, the authors should describe the type of damage they refer to and whether it is inflicted upon SGNs and/or hair cells.

We have amended these sentences to clarify our focus on the auditory system: “The spiral ganglion neurons constitute the primary connection between auditory hair cells and the brain. The spiral ganglion afferent fibers and their synapse with hair cells do not regenerate to any significant degree in adult ears after damage.”

Future studies involving peripheral vestibular organs are beyond the scope of the current paper, and have been the subject of many papers, including several recent ones:

Wang T, Niwa M, Sayyid ZN, et al. Uncoordinated Maturation of Developing and Regenerating Postnatal Mammalian Vestibular Hair Cells. Vol 17.; 2019. doi:10.1371/journal.pbio.3000326

Sayyid ZN, Wang T, Chen L, Jones SM, Cheng AG. Atoh1 Directs Regeneration and Functional Recovery of the Mature Mouse Vestibular System. Cell Rep. 2019;28(2):312-324.e4. doi:10.1016/j.celrep.2019.06.028

Travo C, Gaboyard-Niay S, Chabbert C. Plasticity of Scarpa’s ganglion neurons as a possible basis for functional restoration within vestibular endorgans. Front Neurol. 2012;JUN(June):1-10. doi:10.3389/fneur.2012.00091

Lines 46-49. References to support claims made in the first 3-4 sentences are lacking.

Regeneration of auditory neurons and their peripheral fibers would be clinically significant because auditory nerve and hair cell synaptic dysfunction often accompanies hearing loss (Rask-Andersen and Liu, 2015; Kujawa and Liberman, 2009; Wu et al, 2019). The primary afferent neurons of the auditory system are postsynaptic to sensory hair cells.

Rask-Andersen H, Liu W. Auditory nerve preservation and regeneration in man: Relevance for cochlear implantation. Neural Regen Res. 2015;10(5):710-712. doi:10.4103/1673-5374.156963

Kujawa SG, Liberman MC. Adding Insult to Injury: Cochlear Nerve Degeneration after “Temporary” Noise-Induced Hearing Loss. J Neurosci. 2009;29(45):14077-14085. doi:10.1523/JNEUROSCI.2845-09.2009

Wu PZ, Liberman LD, Bennett K, de Gruttola V, O’Malley JT, Liberman MC. Primary Neural Degeneration in the Human Cochlea: Evidence for Hidden Hearing Loss in the Aging Ear. Neuroscience. 2019;407:8-20. doi:10.1016/j.neuroscience.2018.07.053

Line 56. It would be helpful to add “in maturity” to end of sentence “…..terminals showed no evidence for synaptic recovery”.

We have changed the sentence to “…neurons after noise-induced loss of terminals showed no evidence for synaptic recovery in mature animals1.”

Line 66: The authors should define kainite when first introduced.

The revised manuscript defines kainate as a neuroexcitatory glutamate analogue that activates glutamate receptors. Since glutamate is the principal excitatory neurotransmitter in the cochlea, kainate is used to model glutamate excitotoxicity ex vivo (Wang, 2011) and in vivo (Pujol, 1985).

Wang Q, Green SH. Functional Role of Neurotrophin-3 in Synapse Regeneration by Spiral Ganglion Neurons on Inner Hair Cells after Excitotoxic Trauma In Vitro. J Neurosci. 2011;31(21):7938-7949. doi:10.1523/jneurosci.1434-10.2011

Pujol R, Lenoir M, Robertson D, Eybalin M, Johnstone BM. Kainic acid selectively alters auditory dendrites connected with cochlear inner hair cells. Hear Res. 1985;18(2):145-151. doi:10.1016/0378-5955(85)90006-1

Line 71: This is a transition back to the current study, and this is not clear as written. I suggest you add “In the model we used in this study..”.

The sentence has been amended to read “In the model we used in this study, derived from a newborn animal, the fibers regrow and form synapses with hair cells.”

Line 72. It is not clear what the authors mean by “reinnervation is approximately 50%”? Only 50% of fibers return? Only 50% of synapses return? Are there any clues as to why only 50%?

The revised manuscript clarifies that 60 – 70 % of auditory nerve fibers return at 24h, as quantified in new Fig. 1D.

Lines 74-76. These two sentences starting at “Comparison to…” seem out of place and would be more helpful at the end of page 3 (i.e., around lines 68-70).

We have moved up one of these sentences to clarify this point

Line 79: “Elucidates” seems like too strong a word – perhaps “explores” or “begins to characterize”?

We have amended the sentence to read “begins to characterize”.

Line 106. What fraction of the tissue collected and analyzed comprises the OC, which is defined as the cell population sitting on the basilar membrane? It seems disingenuous to simply refer to this tissue as OC because other cells and tissues are harvested. It would also be helpful to the reader to describe the cell types of the SGN that are isolated (glia, perineural connective tissue, endothelial cells, etc).

The revised Fig. 1 now includes a new panel B that specifies where the cut was made to separate the “OC” fraction from the “SG” fraction. The OC fraction includes the organ of Corti and adjacent supporting cells. The SG fraction includes SGNs and surrounding Schwann cells, perineural connective tissue and cells of the spiral limbus.

Lines 125-129: Is it possible to estimate how many genes, or what % of the genome, are represented?

We have included “Illumina arrays provide a detection p-value (0=max confidence in a gene being detected) for each of the probes. If we use 0 as a threshold to call a gene present, 11,991/25,697 (47%) probes are called present in at least one sample in the organ of Corti, and 12,165/25,697 (47%) in the spiral ganglion tissue.“ in the revised manuscript (lines 133 – 136).

Line 148-149. What is the cutoff they used for “expressed” for a given gene?

 We did not subset probes based on expression. Low-expressed genes are typically more variable in their measurements and therefore larger changes across conditions are needed to achieve statistical significance.

Line 208. Is 8) a typo?

We have corrected the typo

Line 234-36. In this first sentence, the authors should specify they are discussing “peripheral” terminal processes in, I presume, the rat.

We have clarified this in the revised manuscript.

Line 281-300 and elsewhere: The molecular terminology is confusing. I recommend spelling out the full gene/protein then providing the abbreviation for every gene when it is first introduced. It is helpful to have both protein and DNA names when comparing to data. Also, this same section is quite long, with some of the content more suited to the Discussion.

We agree with the reviewer and have revised the paragraph and moved it to appropriate sections of the discussion.

Line 284: References are lacking for RGM’s function in nerve regeneration or development. Is there evidence that RGMs inhibit neurite regeneration in other tissues? If so, this should be described.

We have changed “an axonal growth inhibitor” to “… modulator” and added a reference describing how RGMb promotes neurite outgrowth in sensory dorsal root ganglia (Ma, 2011):

Ma CHE, Brenner GJ, Omura T, et al. The BMP Coreceptor RGMb Promotes While the Endogenous BMP Antagonist Noggin Reduces Neurite Outgrowth and Peripheral Nerve Regeneration by Modulating BMP Signaling. J Neurosci. 2011;31(50):18391-18400. doi:10.1523/JNEUROSCI.4550-11.2011

Line 290. Sema7a is “a link” between inflammation and regeneration, not “the link”.

We have corrected this typo.

Lines 324-355. This is a long and complicated section - more so than it needs to be. A few things that might help make it easier to navigate: break it up into smaller paragraphs; describe the full gene name, the family of proteins it encodes, and known function of all genes found in the network before describing their relationships in the networks; move discussion elements (hypotheses about functional significance of relationships in the network, etc.) to Discussion.

We have amended this section as recommended by the reviewer.

Line 359 – missing figure number.

We have corrected this typo.

Line 372 – “can be achieved” seems like it is something the experimenter is accomplishing, while the mouse is actually doing so. How about “is induced”?

We have implemented this suggestion.

Line 386 - What is the significance of this temporal difference? What type of genes are differentially expressed in each group, and what might that mean?

We believe the tissue specific action of kainate on neurons is reflected by the higher degree of differentially expressed genes in the spiral ganglion tissue and that gene expression is driven by neural regeneration.

Line 391. How do the authors know that the swelling and bursting of the neural endings are the triggers for changes in gene expression? Perhaps it was other cellular changes that occurred in response to kainate treatment? Do they have any ideas about which cell types underwent most gene changes - neurons, glia?

With the microarray technique that we employed, we do not know which cell types’ genes are differentially expressed because the tissues we studied (i.e. OC vs SGN) include multiple cell types. Our results however strongly motivate a future study that would use single-cell RNAseq to address the important and interesting question raised by the reviewer. While that study is beyond the scope of the current paper, we have amended the conclusion of our manuscript to “Our results provide insight into candidates that could be targeted to enhance regeneration in diseased ears in the future and motivate a future study based on single-cell RNAseq to identify specific cell types that undergo changes in gene expression.”

Line 397 – The statement that “the first phase after kainate treatment involves cell migration and chemotaxis and the second involves neuronal regeneration” is simplistic. E.g., it is most likely that no cells in this system were “migrating”. The authors probably mean to say that genes associated with these cellular processes were upregulated, but this does not indicate that these processes are occurring. Indeed, the functions of many genes in a given context are unknown and need to be tested.

We agree with the reviewer and have amended the manuscript to indicate that future studies are needed to determine the functions of many genes that drive transcriptional changes we have observed in this study.

Why do the authors surmise that the increase in inflammatory genes is related to regeneration in this study rather than injury? Also, although they discuss the positive effects of inflammation upon regeneration, there are also many papers describing the adverse influences of inflammation on cellular or nerve regeneration. These should be discussed too.

We agree with the reviewer and have amended the manuscript accordingly.

Line 419. The authors should provide references for Ntrk3.

We have added following reference:

Green SH, Bailey E, Wang Q, Davis RL. The Trk A, B, C’s of Neurotrophins in the Cochlea. Anat Rec. 2012;295(11):1877-1895. doi:10.1002/ar.22587

Line 422: What does “Link neurogenesis to inflammation” mean? Have Sema7a or Syt8 been examined in the cochlea? This sentence starting with “By contrast…” doesnt make any sense here.

We have changed “neurogenesis” to “neural regeneration” and provided a reference:

Namavari A, Chaudhary S, Ozturk O, et al. Semaphorin 7a Links Nerve Regeneration and Inflammation in the Cornea. Investig Opthalmology Vis Sci. 2012;53(8):4575. doi:10.1167/iovs.12-9760

Line 426. What is “part of a phoshatase” mean?

We have corrected the typo to “part of a phosphatase”.

Generally, the Discussion is long, unfocused, and redundant (e.g., a given gene is discussed in several places). It would be helpful if the authors were to condense this section into 2-3 main sections that succinctly interpret the findings.

Also, some interpretations of the findings are simplistic here. Even though the function of specific genes is implicated in ontology analysis or by past papers, it does not mean that these genes are driving these processes in this context. For instance, the idea that Sox21 upregulation in the P5 OC reflects new neurogenesis is illogical. Nerve regeneration in the SGN has several cellular phases - damage, survival, neurite outgrowth, pruning, synaptogenesis, etc. - none of which include neurogenesis.

We thank the reviewed for this constructive criticism and have amended the manuscript accordingly to add several relevant references:

Matsuda S, Kuwako K ichiro, Okano HJ, et al. Sox21 promotes hippocampal adult neurogenesis via the transcriptional repression of the Hes5 gene. J Neurosci. 2012;32(36):12543-12557. doi:10.1523/JNEUROSCI.5803-11.2012

Whittington N, Cunningham D, Le TK, De Maria D, Silva EM. Sox21 regulates the progression of neuronal differentiation in a dose-dependent manner. Dev Biol. 2015;397(2):237-247. doi:10.1016/j.ydbio.2014.11.012

Line 458. This section says two apparently contradictory things: the change in Ntrk3 is specific to regeneration but also may promote neuronal survival. These are two different processes.

We have added 2 references on Ntrk3’s (TrkC) key role in regeneration and survival.

Huang EJ, Reichardt LF. Trk Receptors: Roles in Neuronal Signal Transduction. Annu Rev Biochem. 2003;72(1):609-642. doi:10.1146/annurev.biochem.72.121801.161629

Green SH, Bailey E, Wang Q, Davis RL. The Trk A, B, C’s of Neurotrophins in the Cochlea. Anat Rec. 2012;295(11):1877-1895. doi:10.1002/ar.22587

Figure 2: I suspect the resolution of the text in this figure will be too poor in the final printed version.

The amended manuscript includes a new supplemental Table 1 that lists all sample names included along the axes.

Figure 3: It is unclear what the significance of the pink and green regions are in B.

We apologize for not making this clearer. For each overrepresented Gene Ontology category, in green is the proportion of downregulated differentially expressed genes; in red the proportion of upregulated genes). For example, 6 genes in the defense response GO category are more than expected by chance (p<0.05), therefore overrepresented, and they are all downregulated by kainite treatment in SG at 5 hours. We have added this information to the Figure legend.

Figure 4. Graphs are very hard to understand. What is represented by each dot? What are the red arrows showing? I cannot tell which genes are significantly different between control and treated. It would be helpful if the authors added X axis labels for SGN (5h, 24h, 72h) and OC (5h, 24h, 72h) and arrange data as a column or dot graph with error bars, adding lines with asterisks to indicate which groups have significant difference relative to control.

We thank the reviewer for this comment and have improved the plots. We have separated SG and OC, superimposed a boxplot, eliminated colors, and labeled the x axis.

Re. Figure 5. The authors do not describe the rationale for identifying genes that overlap in 2 or more groups. Why did they make this choice, and what is the potential functional significance of genes that are altered at both timepoints?

The amended manuscript clarifies that we have focused on genes that overlap in two or more groups because these genes are active throughout the regeneration process. (Line 308)

---

## [Editor Report · Decision Letter 1]

20 Aug 2020

Altered Expression of Genes Regulating Inflammation and Synaptogenesis during Regrowth of Afferent Neurons to Cochlear Hair Cells

PONE-D-20-10023R1

Dear Dr. Stankovic,

We’re pleased to inform you that your manuscript has been judged scientifically suitable for publication and will be formally accepted for publication once it meets all outstanding technical requirements.  Thank you for your careful attention to the comments of the reviewers.  

Kind regards,

Clifford R. Hume, MD PHD

Academic Editor

PLOS ONE
---

## [Editor Report · Acceptance letter]

22 Sep 2020

PONE-D-20-10023R1

Altered Expression of Genes Regulating Inflammation and Synaptogenesis during Regrowth of Afferent Neurons to Cochlear Hair Cells

Dear Dr. Stankovic:

I'm pleased to inform you that your manuscript has been deemed suitable for publication in PLOS ONE. Congratulations! Your manuscript is now with our production department.

Kind regards,

on behalf of

Dr. Clifford R. Hume 

Academic Editor

PLOS ONE